# Eating disorder treatment in routine clinical care: A descriptive study examining treatment characteristics and short-term treatment outcomes among patients with anorexia nervosa and bulimia nervosa in Germany and Switzerland

Kathrin Schopf[1]*, Silvia Schneider[1], Andrea Hans Meyer[2], Julia Lennertz[3], Nadine Humbel[2], Nadine-Messerli Bürgy[2,4], Andrea Wyssen[5], Esther Biedert[2], Bettina Isenschmid[6], Gabriella Milos[7], Malte Claussen[8], Stephan Trier[9], Katherina Whinyates[10], Dirk Adolph[1], Tobias Teismann[1], Jürgen Margraf[1], Hans-Jörg Assion[11], Bianca Überberg[11], Georg Juckel[12], Judith Müller[13], Benedikt Klauke[13], Simone Munsch[2]

1 Mental Health Research and Treatment Center, Ruhr University Bochum, Bochum, Germany, 2 Department of Psychology, Clinical Psychology and Psychotherapy, University of Fribourg, Fribourg, Switzerland, 3 Praxis für Psychotherapie, Dortmund, Germany, 4 Institute of Psychology, Clinical Child and Adolescent Psychology, University of Lausanne, Lausanne, Switzerland, 5 University Hospital of Child and Adolescent Psychiatry and Psychotherapy, University of Bern, Bern, Switzerland, 6 Center for Eating Disorders and Obesity, Clinic Zofingen, Zofingen, Switzerland, 7 Department of Consultation-Liaison Psychiatry and Psychosomatic Medicine, University Hospital, Zurich, Switzerland, 8 Department of Psychiatry, Psychotherapy and Psychosomatics, Psychiatric University Hospital Zurich, University of Zurich, Zurich, Switzerland, 9 Privat Clinic Aadorf, Aadorf, Switzerland, 10 Privat Clinic Schützen Rheinfelden, Rheinfelden, Switzerland, 11 Department of Psychiatry, Psychotherapy and Psychosomatic Medicine, LWL-Klinik Dortmund, Dortmund, Germany, 12 Department of Psychiatry, Psychotherapy and Preventive Medicine, LWL University Hospital Bochum, Ruhr University Bochum, Bochum, Germany, 13 Christoph-Dornier-Klinik for Psychotherapy, Münster, Germany

* kathrin.schopf@rub.de

## Abstract

This descriptive study examined patient characteristics, treatment characteristics, and short-term outcomes among patients with Anorexia Nervosa (AN) and Bulimia Nervosa (BN) in routine clinical care. Results for patients receiving full-time treatment were contrasted with results for patients receiving ambulatory treatment. Data of a clinical trial including 116 female patients (18–35 years) diagnosed with AN or BN were subjected to secondary analyses. Patients were voluntarily admitted to one of nine treatment facilities in Germany and Switzerland. Patients received cognitive-behavioral interventions in accordance with the national clinical practice guidelines for the treatment of EDs under routine clinical care conditions, either as full-time treatment or ambulatory treatment. Assessments were conducted after admission and three months later. Assessments included a clinician-administered diagnostic interview (DIPS), body-mass-index (BMI), ED pathology (EDE-Q), depressive symptoms (BDI-II), symptoms of anxiety (BAI), and somatic symptoms (SOMS). Findings showed that treatment intensity differed largely by setting and site, partly due to

**Data Availability Statement:** All relevant data are within the manuscript and its Supporting Information files.

**Funding:** This work was supported by the German Research Foundation (recipient: SSc, Grant SCHN 415/4-1, www.dfg.de) the Swiss National Science Foundation (recipient: SM, Grant 100013:149416, www.snf.ch) and the Swiss Anorexia Nervosa Foundation (recipient: SM, Grant 22-12, www. anorexia-nervosa.ch). We acknowledge support by the DFG Open Access Publication Funds of the Ruhr-Universität Bochum. None of the funders had a role in the study design, collection, analysis, or interpretation of the data, writing the manuscript, or the decision to submit the paper for publication.

**Competing interests:** The authors have declared that no competing interests exist.

national health insurance policies. Patients with AN in full-time treatment received on average 65 psychotherapeutic sessions and patients with BN in full-time treatment received on average 38 sessions within three months. In comparison, patients with AN or BN in ambulatory treatment received 8–9 sessions within the same time. Full-time treatment was associated with substantial improvements on all measured variables for both women with AN ($d = .48$-$.83$) and BN ($d = .48$-$.81$). Despite the relatively small amount of psychotherapeutic sessions, ambulatory treatment was associated with small increases in BMI ($d = .37$) among women with AN and small improvements on all measured variables among women with BN ($d = .27$-$.43$). For women with AN, reduction in ED pathology were positively related to the number of psychotherapeutic sessions received. Regardless of diagnosis and treatment setting, full recovery of symptoms was rarely achieved within three months (recovery rates ranged between 0 and 4.4%). The present study shows that a considerable amount of patients with EDs improved after CBT-based ED treatment in routine clinical care within three months after admission. Intensive full-time treatment may be particularly effective in quickly improving ED-related pathology, although full remission of symptoms is typically not achieved. A small amount of ambulatory sessions may already produce considerable improvements in BN pathology and weight gain among women with AN. As patient characteristics and treatment intensity differed largely between settings, results should not be interpreted as superiority of one treatment setting over another. Furthermore, this study shows that treatment intensity is quite heterogeneous, indicating the possibility for increasing effectiveness in the treatment of EDs in routine clinical care.

## Introduction

Eating disorders (EDs) including Anorexia nervosa (AN) and Bulimia nervosa (BN) are severe psychosomatic disorders with lifetime prevalence among females ranging from 2–4% [1, 2]. EDs are associated with psychological distress, functional impairment, high levels of comorbidity, poor quality of life, medical complications, and increased mortality rates [3–6]. Cognitive-behavioral interventions constitute an evidence-based treatment approach to EDs [7–9]. While clinical trials typically adhere to stringent treatment protocols, treatment programs in routine clinical care are often heterogeneous in terms of treatment components, treatment length, and treatment intensity [10, 11]. To assist practitioners and to improve the quality of care provided to patients in clinical practice, clinical guidelines are becoming increasingly important [12]. However, research examining the implementation of clinical guidelines in routine clinical practice is scarce.

EDs can be treated in a variety of treatment settings. Different treatment settings offer varying degrees of treatment intensity and treatment structure, which may influence treatment effectiveness and costs. However, there is little empirical guidance as to what is the most appropriate treatment setting for an individual patient [13]. Treatment settings include full-time treatment (including overnight stays), which typically means admission to a medical or psychiatric hospital or a residential treatment facility with a multidisciplinary treatment program and high levels of care and restrictions. Day patient treatment (e.g., part-time hospitalization) also offers a multidisciplinary approach as well as a highly-structured treatment but without overnight stays. Ambulatory treatment typically involves one or two sessions per week with a therapist of a single discipline, preferably supported by medical monitoring, and constitutes the least restrictive and usually the least costly treatment.

Several clinical guidelines exist to guide treatment decisions for EDs [14]. According to the American Psychology Association (APA) Practice Guideline [7], hospitalization of patients with EDs is vital in case of acute medical complications. Also, severe forms of EDs, psychological comorbidity, environmental stressors, or unavailability of other treatment options may also warrant hospitalization of patients. However, it is unclear if normalization of eating behavior and weight restoration can be best achieved in a hospital setting after acute medical instabilities have been addressed. There is emerging evidence that less-intensive treatment approaches (e.g., day patient or ambulatory treatment) achieve similar outcomes as hospitalization among medically stable patients with EDs [15, 16]. The British guideline of the National Institute for Health and Clinical Excellence (NICE) even suggests that the majority of patients with AN and BN should be treated in an ambulatory setting [8]. The German Clinical Practice Guideline [17] states that hospitalization is often necessary for patients suffering from AN, while ambulatory treatment is the treatment of choice for patients suffering from BN. The authors state that an evidence-based decision regarding the most appropriate treatment setting is limited by the small amount of empirical data and treatment decisions are often based on clinical experience and expert opinions.

As the results of randomized controlled trials (RCTs) may not always directly map the real-world situation, more outcome research in routine clinical care is needed [11, 16, 18–22]. RCTs are typically conducted under optimal conditions (e.g., selected patients, trained and supervised therapists, adherence to treatment manual), which are different from those found in routine clinical practice. Although observational studies in routine clinical practice typically lack a control group (which impedes causal conclusions regarding treatment effectiveness), they may provide valuable information regarding treatment characteristics and outcomes in public health care [23].

The present study aimed to examine patient characteristics, treatment characteristics, and short-term outcomes among women suffering from AN or BN, who were treated with CBT-based interventions according to national clinical practice treatment guidelines in nine different treatment institutions in Germany and Switzerland. The following research questions were examined: What are the differences between patients admitted to full-time treatment and patients admitted to ambulatory treatment? How much treatment and which treatment components do patients with EDs typically receive within three months after treatment initiation? Are there differences between full-time treatment and ambulatory treatment? What are short-term outcomes of guideline-based cognitive-behavioral interventions conducted in routine clinical care among patients with EDs? Which variables are associated with symptomatic improvements?

## Materials and methods

### Participants and procedure

Data were drawn from a large clinical trial [24, 25] in which behavioural, emotional, and physiological responses to mass media exposure were compared between women with EDs, women with other mental disorders, and women without mental disorders before and after guideline-oriented treatment. The study was approved by the ethical committee of the Faculty of Psychology at the Ruhr University Bochum in Germany (reference no. 142) and the local Ethics Research Committee of the canton of Fribourg in Switzerland (reference no. 023/12-CER-FR). Participation was voluntary and each participant provided informed written consent. The study was registered in the German Clinical Trials Registry (trial number: DRKS00005709). For the present study, only data of patients with a primary diagnosis of AN or BN were used. An overview of the study is provided in Fig 1.

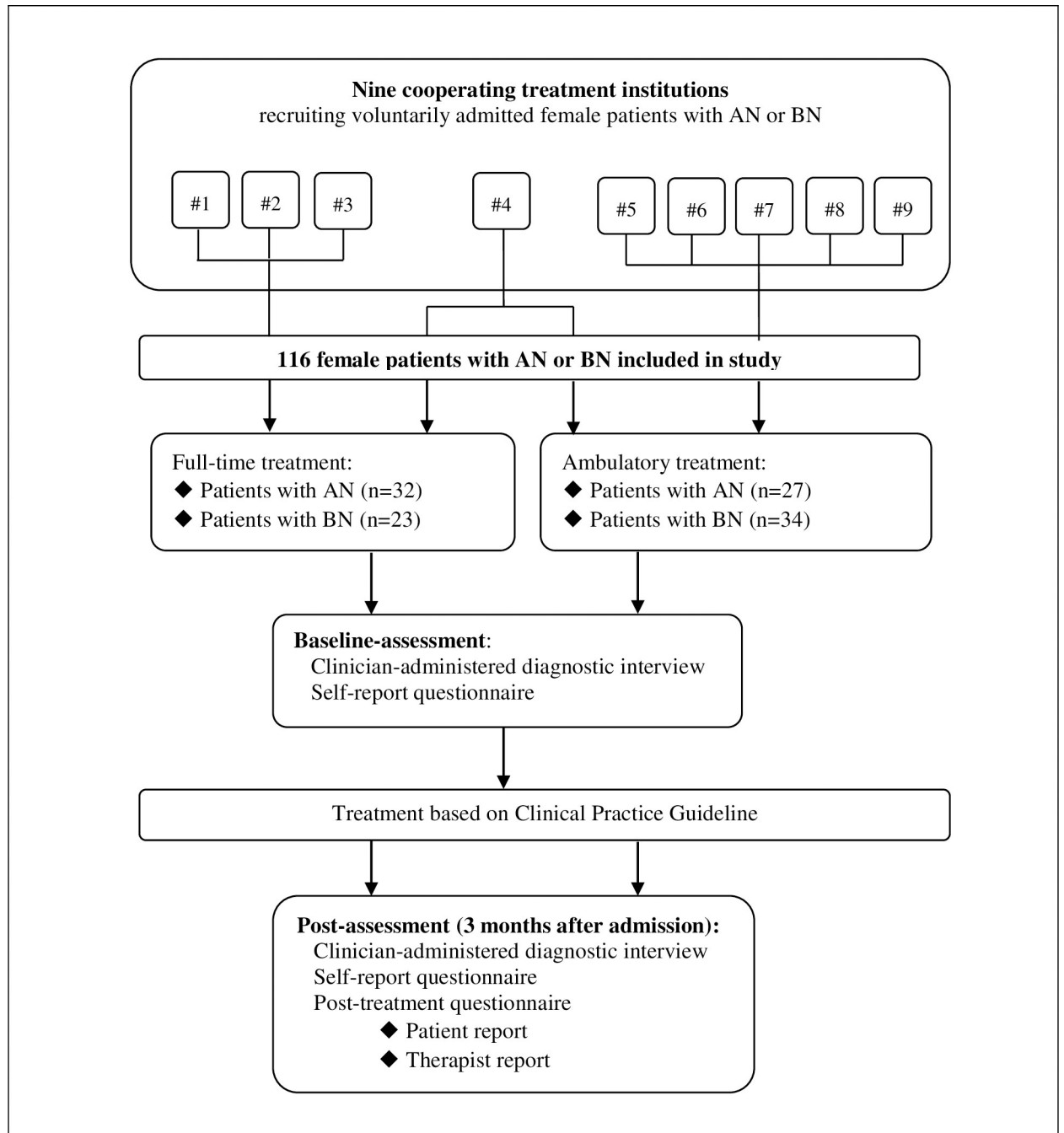

**Fig 1. Overview of study.**

Sixty-four patients with AN and 66 patients with BN were enrolled in the study. Fifty-nine patients with AN and 57 patients with BN participated. Potentially eligible patients who were voluntarily admitted between July 2014 and April 2017 to cooperating treatment centers were approached by staff members and informed about the clinical study. Study eligibility criteria included female gender, age between 18 and 35 years, and a primary diagnosis of AN or BN. Exclusion criteria were current pregnancy, breast feeding, or intake of beta-blockers. Patients

were assessed after treatment admission and three months later. Assessments included a clinician-administered diagnostic interview (60–90 minutes) and a self-report questionnaire (60 minutes).

## Cooperating sites and treatment

The study sites included nine treatment sites in western Germany and northern Switzerland, which provided CBT-based interventions in accordance with the national clinical practice treatment guideline for the treatment of ED [17]. Sixty-three patients (54.3%) were recruited in Swiss treatment sites and 53 (45.7%) in German treatment sites. Three sites provided ED treatment as full-time treatment, one side provided ED treatment as either full-time or ambulatory treatment, and five sites provided ED treatment as ambulatory treatment. Sixty-one patients (52.6%) received outpatient treatment, 42 (36.2%) received inpatient treatment, and 13 (11.2%) received residential treatment.

Full-time treatment consisted of 24/7-care in a medical or psychiatric hospital or a privately governed non-hospital treatment facility, provided by a multidisciplinary team, including individual and group therapy, nutritional counselling, meal support, and occupational/recreation therapy. Ambulatory treatment typically consisted of one or two sessions of individual psychotherapy per week provided by a psychotherapist. It should be noted that the number of treatment sessions for ambulatory patients was restricted by procedural standards of the German health care system. At the time of the study, health insurance companies in Germany only reimbursed a maximum of five ambulatory treatment sessions (probatory/diagnostic sessions). To continue treatment, health insurance policies required a report, which was evaluated in a peer review process before approval of the reimbursement of treatment costs was granted. This procedure typically resulted in a treatment break of 4–6 weeks, with no contact between the patient and the therapist. Patients' insurance companies covered the treatment costs and admission to treatment was voluntarily.

Treatments in all sites were conducted as usual. All sites confirmed that their treatment standards adhere to current clinical practice guidelines for the treatment of ED [17]. The guidelines include psychoeducation about ED, nutritional counselling, analyses of individual problem behaviors and goals, development of an individual model to understand the development and maintenance of the disorder, normalization of eating behavior and restitution of a normal body weight, stimulus control and response control, improvement of body image, improvement of interpersonal skills, reduction of interpersonal conflicts, and relapse prevention.

## Measures

**Diagnostic interview.** A clinician-administered, semi-structured diagnostic interview (DIPS; Diagnostic Interview for Mental Disorders) [26] was used to determine the presence or absence of any mental disorders at pre- and post-measurement. The DIPS is based on the DSM-IV-TR [27] with good psychometric properties [28]. For the purpose of our study, the DIPS EDs section was adapted according to DSM-5 [29]. In addition to clinical diagnoses, ED severity (i.e., mild, moderate, severe, extreme) based on DSM-5 [29] criteria was also specified. Furthermore, the presence or absence of a comorbid disorder was coded (no, yes). Interviews were conducted at the treatment site or by telephone (if patients were discharged). All interviewers were trained and supervised. Interviews were recorded and 10% of the interviews were additionally coded by two independent raters. Interrater reliability for primary diagnoses (Fleiss K = .85) and comorbid diagnoses (Fleiss K = .80) was good.

**Self-report questionnaire.** At pre- and post-measurement, all patients completed an online questionnaire including questions regarding age, education, nationality, relationships status, use of psychopharmaceutics, and number of cigarettes smoked per day. Symptoms of ED pathology were assessed by the global score of the Eating Disorder Examination Questionnaire (EDE-Q) [30]. Internal consistency is good and convergent as well as divergent validity have been demonstrated [31]. Depressive symptoms were assessed using the 21-item Beck Depression Inventory-2 (BDI-II) [32]. Good internal consistency, test-retest reliability, sensitivity and specificity as well as clinical utility for detecting depression have been shown [33]. Symptoms of anxiety were assessed using the 21-item Beck Anxiety Inventory (BAI) [34]. The BAI has demonstrated high internal consistency, acceptable test-retest reliability, and acceptable discriminant and convergent validity [35]. Somatoform symptoms were assessed using the female 52-item version of the Screening for Somatoform Symptoms (SOMS) [36]. The scale shows high internal consistency (alpha = .92) and validity has been established [36].

**Body-Mass-Index (BMI).** At pre- and post-measurement, weight and height of study participants were measured using a calibrated electronic scale (Seca 899, Basel, Switzerland) and a stadiometer (Seca 213, Basel, Switzerland). BMI was calculated as weight in kilograms divided by height in meters squared.

**Remission rates.** Patients were considered fully remitted if they met the following criteria: a) Full remission based on diagnostic interview (i.e., after full criteria for AN/BN were previously met, none of the criteria have been met for a sustained period of time), b) BMI $\geq$ 18.5 and c) EDE-Q global score $\leq$ 2.77. Patients were considered partially remitted if they met the following criteria: Partial remission based on diagnostic interview (i.e., for AN: after full criteria for AN were previously met, criterion A (low body weight) has not been met for a sustained period of time, but either criterion B (intense fear of gaining weight or becoming fat or behavior that interferes with weight gain) or criterion C (disturbances in self-perception of weight and shape) is still met; for BN: after full criteria for BN were previously met, some but not all, of the criteria have been met for a sustained period of time) and b) BMI $\geq$ 18.5. Remission rate criteria were based on definitions of previous studies [37–39].

**Treatment intensity and treatment characteristics.** To examine treatment intensity and treatment characteristics, brief online questionnaires were administered to therapists at post-measurement. Items assessed the number of individual and group sessions as well as treatment components and treatment goals discussed with patients (a list of response options was provided). The response rate of therapists was 67%. Patients received a similar brief online questionnaire. The response rate of patients was 77%.

## Statistical analyses

Statistical analyses were conducted using SPSS 27 (IBM). To describe the total sample as well as subsamples by diagnosis (AN vs. BN) and setting (full-time treatment vs. ambulatory treatment), means, standard deviations, frequencies, and percentages of measured variables were reported. Statistical comparisons between groups on several related continuous variables were based on multivariate analyses of variance. Statistical comparisons of categorical variables were based on chi-square tests. All comparisons were made at the level of $p < .05$. To examine symptomatic changes on ED-related variables, Cohen's *d* was calculated as the mean difference between pre- and post-measurement scores divided by the pooled standard deviation. To examine variables related to symptomatic improvements, we used a stepwise procedure. Potential correlates of symptomatic improvements (as reported in Tables 1 and 4) were first examined using Pearson correlations. Only statistically significant variables were then entered into multivariate regression model. Dependent variables were

**Table 1. Descriptive statistics at baseline (treatment admission) by diagnosis and treatment setting.**

| Characteristic | Patients with AN | | | Patients with BN | | |
|---|---|---|---|---|---|---|
| | Total AN sample (n = 59) | Ambulatory treatment (n = 27) | Full-time treatment (n = 32) | Total BN sample (n = 57) | Ambulatory treatment (n = 34) | Full-time treatment (n = 23) |
| Age (M, SD) | 22.4 (4.2) | **24.0 (4.5)** [b] | **21.3 (3.4)** [b] | 23.1 (4.1) | 22.9 (3.7) | 23.5 (4.7) |
| Nationality % (n) | | | | | | |
| German | 50.0 (29) | 63.0 (17) | 38.7 (12) | 50.0 (27) | **69.7 (23)** [b] | **19.0 (4)** [b] |
| Swiss | 48.3 (28) | 33.3 (9) | 61.3 (19) | 46.3 (25) | **24.2 (8)** [b] | **81.0 (17)** [b] |
| Other | 1.7 (1) | 3.7 (1) | 0.0 (0) | 3.7 (2) | **6.1 (2)** [b] | **0.0 (0)** [b] |
| Education (%, n) | | | | | | |
| Still in school | 3.4 (2) | 0.0 (0) | 6.5 (2) | 1.9 (1) | 3.1 (1) | 0.0.(0) |
| Compulsory education | 16.9 (10) | 22.2 (6) | 12.9 (4) | 15.1 (8) | 15.6 (5) | 14.3 (3) |
| HS diploma or job training | 57.6 (34) | 48.1 (13) | 67.7 (21) | 67.9 (36) | 68.8 (22) | 66.7 (14) |
| University degree | 20.3 (12) | 29.6 (8) | 12.9 (4) | 15.1 (8) | 12.5 (4) | 19.0 (4) |
| Relationship status % (n) | | | | | | |
| Not in a relation | 56.9 (33) | 51.9 (14) | 61.3 (19) | 51.9 (28) | 51.5 (17) | 52.4 (11) |
| In a relation/ married | 43.1 (25) | 84.1 (13) | 38.7 (12) | 46.3 (25) | 48.5 (16) | 42.8 (9) |
| BMI (M, SD) | **17.2 (1.6)** [a] | **17.7 (1.5)** [b] | **16.7 (1.6)** [b] | **22.7 (2.5)** [a] | 23.1 (2.8) | 22.1 (2.0) |
| EDE-Q score (M, SD) | *3.8 (1.2)* [c] | 3.9 (1.3) | 3.4 (1.1) | *4.2 (1.2)* [c] | 4.2 (1.3) | 4.2 (1.1) |
| BDI score (M, SD) | 25.4 (11.8) | 23.0 (12.2) | 25.9 (10.5) | 26.4 (11.7) | 25.9 (11.7) | 26.5 (11.9) |
| BAI score (M, SD) | 20.6 (11.5) | 19.7 (11.7) | 20.0 (11.1) | 19.6 (8.9) | 18.1 (8.8) | 21.6 (9.0) |
| SOMS score (M, SD) | 11.4 (8.2) | 11.7 (7.8) | 11.9 (10.3) | 10.0 (6.6) | 10.4 (6.9) | 11.9 (10.3) |
| Use of psychopharmaceutics % (n) | | | | | | |
| Yes | **23.7 (14)** [a] | 14.8 (4) | 31.3 (10) | **38.6 (22)** [a] | **23.5 (8)** [b] | **60.9 (14)** [b] |
| Comorbidity % (n) | | | | | | |
| Yes | 49.2 (29) | 37.0 (10) | 59.4 (19) | 54.4 (31) | **41.2 (14)** [b] | **73.9 (17)** [b] |
| ED severity %, (n) | | | | | | |
| Mild | 28.8 (17) | **51.9 (14)** [b] | **9.4 (3)** [b] | 28.1 (16) | *26.5 (9)* [c] | *30.4 (7)* [c] |
| Moderate | 30.5 (18) | **22.2 (6)** [b] | **37.5 (12)** [b] | 38.6 (22) | *47.1 (16)* [c] | *26.1 (6)* [c] |
| Severe | 27.1 (16) | **14.8 (4)** [b] | **37.5 (12)** [b] | 17.5 (10) | *20.6 (6)* [c] | *13.0 (3)* [c] |
| Extreme | 13.6 (8) | **11.1 (3)** [b] | **15.6 (5)** [b] | 15.8 (9) | *5.9 (2)* [c] | *30.4 (7)* [c] |

Note. AN = Anorexia Nervosa, BN = Bulimia Nervosa, EDE-Q = Eating Disorder Examination Questionnaire, BMI = Body Mass Index, BDI = Beck Depression Inventory, BAI = Beck Anxiety Inventory, SOMS = Somatic Symptoms Scale. ED severity = Eating disorder severity. M = mean, SD = standard deviation, n = sample size.

[a] indicates significant differences between patients with AN and patients with BN (in bold).

[b] indicates differences between patients by treatment setting (in bold).

[c] indicates differences between patients with AN and patients with BN at p = .06 (in italic).

[d] indicates differences between patients by treatment setting at p = .06 (in italic). Percentages constitute valid percentages (with missings being excluded).

symptomatic improvements on primary treatment outcomes, respectively 1) weight gain defined as pre-post BMI difference and 2) ED pathology defined as pre-post EDE-Q difference. Treatment intensity was defined as the sum of individual and group sessions. Analyses were conducted for patients who provided complete data (complete-case analysis). To examine remission rates, intent-to-treat analyses were conducted (missing values were considered not remitted). A total 17 participants (14.7%) dropped out of the study. Drop-out by diagnosis and treatment setting is displayed in Table 5. Reasons were lack of time, study burden, burden of disease, or unavailability.

## Results

### Descriptive characteristics of patients

On average, patients with EDs were 22.8 years ($SD$ = 4.1). A total of 54.3% of patients were Swiss and 45.7% were German. Approximately half of all patients (51.7%) displayed at least one comorbid disorders, most commonly a depressive disorder (38.8%), an anxiety disorder (25.0%), a sleep disorder (6.0%), a somatoform disorder (4.3%), or substance use disorder (2.6%). Descriptive characteristics of patients with AN and BN are displayed in Table 1. Patients with AN differed significantly from patients with BN in BMI ($m$ = 17.2 vs. $m$ = 22.7, respectively) and the use of psychopharmaceutics (23.7% vs. 38.6%, respectively).

Descriptive characteristics of patients between treatment settings are also displayed in Table 1. For patients with AN, there were significant differences between patients receiving ambulatory treatment and patients receiving full-time treatment on age (full-time patients were significantly younger than ambulatory patients, $F$ = 6.3, $p$ = .02), BMI (full-time patients had a significantly lower BMI than ambulatory patients, $F$ = 5.2, $p$ = .03), and ED severity (full-time patients more often displayed a severe or extreme ED severity, while ambulatory patients more often displayed a mild ED severity). For patients with BN, there were significant differences between settings in the use of psychopharmaceutics (full-time patients more often reported the use of psychopharmateucis than ambulatory patients, $\chi^2$ = 8.1, $p$ < .01) the presence of a comorbid disorder (full-time patients were more often diagnosed with a comorbid disorder than ambulatory patients, $\chi^2$ = 5.9, $p$ = .01), and nationality (full-time patients were more often Swiss than ambulatory patients, $\chi^2$ = 16.8, $p$ < .001). Also, there was a marginal significant difference in ED severity (full-time patients more often displayed an extreme ED severity than ambulatory patients, $\chi^2$ = 7.3, $p$ = .06).

### Treatment intensity and treatment characteristics

Table 2 displays treatment components and treatment goals reported by patients and therapists. Table 3 summarizes treatment intensity by diagnosis and treatment setting. Generally, the number of treatment sessions differed greatly between treatment settings. Patients with AN in full-time treatment received on average 28.8 individual sessions plus 36.5 group sessions and patients with BN in full-time treatment received on average 18.7 individual sessions plus 18.9 group sessions within three months after admission. Noteworthy, the range of individual and group sessions differed remarkably between treatment institutions. Patients with AN in ambulatory treatment received on average 9.4 individual sessions and patients with BN in ambulatory treatment received on average 8.1 individual sessions within three months after admission. In the present study, ambulatory patients did not receive any group sessions.

### Symptomatic changes among patients within three months after admission

Table 4 summarizes symptomatic changes on several outcome measures from pre- to post-assessment including corresponding effect sizes (Cohen's $d$). For patients with AN, full-time treatment was associated with large improvements in BMI ($d$ = .83) and moderate to large improvements in somatic symptoms ($d$ = .77), depressive symptoms ($d$ = .77), and symptoms of anxiety ($d$ = .73). Also, moderate improvements were observed in ED pathology ($d$ = .48). For patients with AN, ambulatory treatment was associated with small improvements in BMI ($d$ = .37). For patients with BN, full-time treatment was associated with large improvements in ED pathology ($d$ = .81) and somatic symptoms ($d$ = .85). Also, moderate improvements were observed in depressive symptoms ($d$ = .70) and symptoms of

**Table 2. Treatment components and treatment goals reported by patients and therapists three months after treatment admission.**

| Treatment components | Patients with AN | | Patients with BN | |
|---|---|---|---|---|
| | Ambulatory treatment (n = 22) (n = 20) | Full-time treatment (n = 26) (n = 22) | Ambulatory treatment (n = 24) (n = 22) | Full-time treatment (n = 19) (n = 15) |
| Psychoeducation | | | | |
| Patient report | 77.3 (19) | 96.2 (25) | 66.7 (16) | 89.5 (17) |
| Therapist report | 100.0 (20) | 100.0 (22) | 86.4 (19) | 100.0 (15) |
| Analysis of problems and goals | | | | |
| Patient report | 59.1 (13) | 73.1 (19) | 75.0 (18) | 89.5 (17) |
| Therapist report | 90.0 (18) | 100.0 (22) | 100.0 (22) | 66.7 (10) |
| Definition of disorder-specific goals | | | | |
| Patient report | 72.2 (16) | 92.3 (24) | 66.7 (16) | 84.2 (16) |
| Therapist report | 85.0 (17) | 81.8 (18) | 59.1 (13) | 100.0 (15) |
| Discussion of etiological model | | | | |
| Patient report | 50.0 (11) | 65.4 (17) | 50.0 (12) | 94.7 (18) |
| Therapist report | 90.0 (18) | 95.5 (21) | 95.5 (21) | 100.0 (15) |
| Stimulus-/response control | | | | |
| Patient report | 45.5 (10) | 80.8 (21) | 45.8 (11) | 89.5 (18) |
| Therapist report | 55.0 (11) | 72.7 (16) | 72.7 (16) | 86.7 (13) |
| Improvement of body image/self-perception | | | | |
| Patient report | 31.8 (7) | 46.2 (12) | 12.5 (3) | 42.1 (8) |
| Therapist report | 70.0 (14) | 86.4 (19) | 54.5 (12) | 73.3 (11) |
| Interpersonal conflicts | | | | |
| Patient report | 50.0 (11) | 65.4 (17) | 33.3 (8) | 84.2 (16) |
| Therapist report | 30.0 (6) | 63.6 (14) | 68.2 (15) | 100.0 (15) |
| Relapse prevention | | | | |
| Patient report | 36.4 (8) | 65.4 (17) | 33.3 (8) | 84.2 (16) |
| Therapist report | 60.0 (12) | 100.0 (22) | 68.2 (15) | 100.0 (15) |
| **Treatment goals** | | | | |
| Normalization of body weight* | | | | |
| Patient report | 59.1 (13) | 92.3 (24) | - | - |
| Therapist report | 85.0 (17) | 100.0 (22) | | |
| Normalization of eating behaviour | | | | |
| Patient report | 36.4 (8) | 69.2 (18) | 91.7 (22) | 94.7 (18) |
| Therapist report | 100.0 (20) | 95.5 (21) | 86.4 (19) | 93.3 (14) |
| Treatment of physical symptoms* | | | | |
| Patient report | 54.5 (12) | 65.4 (17) | - | - |
| Therapist report | 45.0 (9) | 54.5 (12) | | |
| Analysis/modification of maintaining factors | | | | |
| Patient report | 86.4 (19) | 80.8 (21) | 70.8 (17) | 94.7 (18) |
| Therapist report | 90.0 (18) | 100.0 (22) | 100.0 (22) | 100.0 (15) |

Note: Treatment may continue after the three-months post-measurement (particularly among ambulatory patients).

* Only assessed among patients with AN. AN = Anorexia Nervosa. BN = Bulimia Nervosa.

anxiety ($d = .48$). For patients with BN, ambulatory treatment was associated with small improvements in ED pathology ($d = .44$), depressive symptoms ($d = .39$), symptoms of anxiety ($d = .31$), and somatic symptoms ($d = .27$).

**Table 3. Descriptive statistics of treatment received within three months after admission (therapist-report).**

| | Patients with AN | | Patients with BN | |
| --- | --- | --- | --- | --- |
| | Ambulatory treatment (n = 19) | Full-time treatment (n = 21) | Ambulatory treatment (n = 21) | Full-time treatment (n = 15) |
| Mean number of individual sessions (range) | 9.4 | 28.8 | 8.1 | 18.7 |
| | (4–23) | (2–84) | (1–16) | (1–48) |
| Mean number of group sessions (range) | 0.0 | 36.5 | 0.0 | 18.9 |
| | (0) | (0–150) | (0) | (1–38) |

Note: Treatment may continue after the three-months (particularly among outpatients). AN = Anorexia Nervosa. BN = Bulimia Nervosa.

## Correlates of symptomatic improvements

Among women with AN, a significant correlation was found only between weight gain and ED severity ($r = .62.$, $p < .001$, $R^2 = .38$), indicatig that a higher AN severity (i.e., more severe underweight) at the time of admission was associated with more weight gain during the first three months of treatment. A reduction in ED pathology was significantly correlated only with treatment intensity ($r = .42.$, $p < .01$, $R^2 = .20$), indicating that more psychotherapeutic sessions were associated with more reduction in ED pathology among women with AN. Among women with BN, no significant correlations could be observed between reduction in ED pathology and any of the variables (all $p > .05$).

## Remission rates among patients within three months after admission

Table 5 summarizes remission rates. Among patients with AN receiving full-time treatment, one (3.1%) was considered fully remitted and six (18.8%) were considered partially remitted

**Table 4. Symptomatic changes (means, standard deviations, Cohen's *d*) among patients three months after treatment admission.**

| | | Patients with AN | | | Patients with BN | | |
| --- | --- | --- | --- | --- | --- | --- | --- |
| | | Total sample (n = 51) | Amb. treatment (n = 24) | Full-time treatment (n = 27) | Total sample (n = 48) | Amb. treatment (n = 29) | Full-time treatment (n = 19) |
| BMI | T1 | 17.1 (1.7) | 17.7 (1.6) | 16.6 (1.6) | 22.7 (2.6) | 23.2 (2.9) | 22.0 (2.0) |
| | T2 | 18.1 (1.5) | 18.3 (1.5) | 18.0 (1.6) | 22.5 (2.7) | 22.9 (3.0) | 22.2 (2.4) |
| | *d* | .60 | .37 | .83 | .08 | .11 | .01 |
| EDE-Q | T1 | 3.7 (1.2) | 3.9 (1.3) | 3.4 (1.1) | 4.2 (1.2) | 4.2 (1.4) | 4.3 (0.9) |
| | T2 | 3.2 (1.4) | 3.7 (1.4) | 2.8 (1.4) | 3.4 (1.6) | 3.5 (1.7) | 3.3 (1.4) |
| | *d* | .32 | .16 | .48 | .58 | .43 | .81 |
| BDI | T1 | 25.0 (11.3) | 23.3 (12.1) | 26.5 (10.6) | 27.2 (11.8) | 26.5 (12.4) | 28.3 (11.0) |
| | T2 | 20.2 (12.3) | 12.6 (13.6) | 18.2 (10.9) | 20.8 (12.8) | 21.5 (12.7) | 19.8 (13.1) |
| | *d* | .40 | .05 | .77 | .52 | .39 | .70 |
| BAI | T1 | 19.5 (10.8) | 19.5 (11.5) | 19.6 (10.4) | 20.0 (9.1) | 18.0 (9.2) | 23.0 (8.2) |
| | T2 | 14.9 (10.9) | 17.5 (12.6) | 12.6 (8.6) | 16.3 (10.7) | 14.9 (10.4) | 18.3 (11.2) |
| | *d* | .43 | .16 | .73 | .37 | .31 | .48 |
| SOMS | T1 | 11.0 (7.6) | 11.6 (8.0) | 10.4 (7.3) | 10.4 (6.7) | 10.6 (7.2) | 9.9 (5.9) |
| | T2 | 7.9 (7.5) | 10.7 (8.0) | 5.2 (6.1) | 7.4 (6.5) | 8.7 (7.1) | 5.4 (4.9) |
| | *d* | .41 | .11 | .77 | .45 | .27 | .85 |

Note. Amb. = Ambulatory, AN = Anorexia Nervosa, BN = Bulimia Nervosa, BMI = Body-Mass-Index, EDE-Q = Eating Disorder Examination Questionnaire, BDI = Beck Depression Inventory, BAI = Beck Anxiety Inventory, SOMS = Somatic Symptoms Scale. M = mean, SD = standard deviation, n = sample size, d = Cohen's d. Descriptive statistics are reported only for patients with complete outcome data (complete-case analysis).

**Table 5. Remission rates at post-measurement (three months after admission).**

| % (n) | Patients with AN | | | Patients with BN | | |
|---|---|---|---|---|---|---|
| | Total sample (n = 59) | Amb. treatment (n = 27) | Full-time treatment (n = 32) | Total sample (n = 57) | Amb. treatment (n = 34) | Full-time treatment (n = 23) |
| **Drop-out** | 11.9 (7) | 11.1 (3) | 12.5 (4) | 17.5 (10) | 17.6 (6) | 17.4 (4) |
| **No remission** | 69.5 (41) | 74.1 (20) | 65.6 (21) | 50.9 (29) | 52.9 (18) | 47.8 (11) |
| **Partial remission** | 16.9 (10) | 14.8 (4) | 18.8 (6) | 28.1 (16) | 26.5 (9) | 30.4 (7) |
| **Full remission** | 1.7 (1) | 0.0 (0) | 3.1 (1) | 3.5 (2) | 2.9 (1) | 4.4 (1) |

Note: AN = Anorexia Nervosa. BN = Bulimia Nervosa, Amb. = Ambulatory.

within three months after admission. Among patients with AN receiving ambulatory treatment, none were considered fully remitted and four (14.8%) were considered partially remitted within three months after admission. Among patients with BN receiving full-time treatment, one (4.4%) was considered fully remitted and seven (30.4%) were considered partially remitted within three months after admission. Among patients with BN receiving ambulatory treatment, one (2.9%) was considered fully remitted and nine (26.5%) were considered partially remitted within three months after admission.

## Discussion

Few empirical data examine the implementation of guideline-oriented CBT for EDs and associated outcomes in routine clinical care. In the present study, findings for full-time treatment (e.g., hospitalization) were contrasted with findings for ambulatory treatment. Across settings, patient characteristics and treatment intensity were quite heterogeneous. Due to these differences, direct comparisons of treatment outcomes between settings are not indicated and results should not be interpreted as superiority of one treatment setting over another. Generally, the present study showed that a substantial number of patients with AN and BN displayed symptomatic improvements on several indicators of mental health within three months of treatment admission, although recovery rates remained quite low.

Among women with AN receiving full-time treatment, substantial improvements were observed in body weight, depressive symptoms, symptoms of anxiety, and somatic symptoms, while improvements in ED pathology were moderate. It should be noted that, particularly for this group, the range of the number of psychotherapeutic sessions received was remarkably broad, indicating strong variations in treatment intensity. Results showed that, for women with AN, reduction in ED pathology was directly related to the number of psychotherapeutic sessions received, while weight gain was associated with higher AN severity (i.e., more severe underweight). Although nearly one fifth of patients with AN was considered partially remitted after three months of full-time treatment, the recovery rate (i.e., full remission of symptoms) was very low (3.1%). In line with this, a multi-centre study of short-term outcomes of inpatient treatment (e.g. hospitalization) similarly reported large increases in BMI and improvements in physical health, but the majority of patients remained in an underweight BMI range and continued to display clinical levels of ED symptoms at discharge [10]. Similarly, Treat and colleagues [40] reported an increase in expected body weight from 71 to 85% in inpatient care. However, one third of patients was discharged against medical advice and not included in the analyses. Among women with AN receiving ambulatory treatment, small improvements in BMI, but no further symptomatic improvements were found, which can be explained, at least in part, by the relatively low number of psychotherapeutic sessions received in the ambulatory

setting compared to the full-time setting. A recent systematic review concluded that individual psychotherapy generally produces good results in patients suffering from AN and that there was no superiority of any specific treatment setting, however weight gains occurred more rapid during hospitalization compared to ambulatory treatment [41]. Another recent systematic review concluded that there were no differences in weight gain among individuals with AN treated in different settings, but patients seemed more likely to complete treatment in settings outside the hospital [13].

For women with BN receiving full-time treatment, large improvements in self-reported ED pathology and somatic symptoms were observed as well as moderate improvements in depressive symptoms and symptoms of anxiety. In line with this, research indicates that nearly half of patients with severe BN showed clinically significant symptom changes after hospitalization [42]. Among women with BN receiving ambulatory treatment, small improvements in ED pathology, depressive symptoms, symptoms of anxiety, and somatic symptoms were observed. Symptomatic changes were unrelated to any of the observed variables, indicating that improvements among BN patients were not specific for patient or treatment characteristics. Noteworthy, remission rates were quite similar for full-time BN patients (full remission: 4.4%, partial remission: 30.4%) and ambulatory BN patients (full remission: 3.1%, partial remission: 26.5%). While outcomes in full-time treatment were somewhat larger, outcomes in ambulatory treatment were achieved with far less resources and a substantially lower number of treatment sessions. The results of the present study underline treatment recommendations of clinical guidelines, which generally recommend ambulatory treatment for individuals suffering from BN and hospitalization only in case of severe forms of BN or associated medical complications [8]. Again, full recovery (i.e., full remission of symptoms) among women with BN was quite rare in both settings (4.4% in full-time treatment, 2.9% in ambulatory treatment) within three months of treatment admission, indicating the need for prolonged treatment even after intensive full-time treatment.

Several differences between the full-time setting and the ambulatory setting are noteworthy. First, as expected, patient characteristics differed between settings. Patients with AN receiving full-time treatment were younger, had a lower BMI, and more often a severe or extreme form of AN compared to ambulatory AN patients. Patients with BN receiving full-time treatment more often reported the use of psychopharmateucis, were more often diagnosed with a comorbid disorder, and more often displayed an extreme form of BN compared to ambulatory BN patients. These findings indicate that ED severity and psychiatric comorbidity are somewhat more severe among patients who voluntarily admitted to full-time treatment compared to ambulatory treatment. Yet, it is noteworthy that approximately half of all full-time AN patients displayed a mild or moderate severity form (average BMI of full-time AN patients was 16.7). This finding indicates that, in clinical practice, hospitalization of patients with AN often occurs for reasons other than very low body weight (i.e., severe form of AN).

Furthermore, treatment intensity differed largely between settings as well as treatment institutions. Full-time patients with AN received on average 65 psychotherapeutic sessions within three months after admission and full-time patients with BN received on average 38 psychotherapeutic sessions. It should be noted, that the range of the number of sessions differed largely, especially among AN patients. In this context, it should be noted that one treatment institution adminstered a highly intensive AN treatment program (including 2–5 individual sessions per day plus daily group sessions, resulting in an average of 130 psychotherapeutic sessions for AN patients in this treatment institution), thereby strongly increasing the average number of psychotherapeutic sessions for full-time AN patients in the present study. In line with this, previous studies have already noted a large variability in treatment programs and outcomes [11]. In comparison, ambulatory patients with AN or BN received 8–9

psychotherapeutic sessions within three months after treatment admission. This number was relatively low, partly due to national health insurance policies at the time of the study (as explained above), which were quite disadvantageous for German outpatients, and may have impeded fast improvements and delayed treatment results. In the light of the relatively small number of treatment sessions, the observed symptomatic improvements among ambulatory patients are particularly noteworthy. Besides large differences in treatment intensity, only small differences were observed in treatment components and treatment goals reported by therapists and patients, which probably relate to the number of treatment sessions.

The present study has several limtations. First of all, generalizability of the present findings may be limited by several factors including small sample size of subgroups, specific sample characteristics of study participants, or specific characteristics of cooperating treatment institutions. It should be acknowledged that both patients and institutions do not represent the general population. Furthermore, the present study only examined a few potential covariates. It is possible that additional patient or treatment characteristics may be associated with treatment setting as well as symptom course. In addition, the present study does not include a control group. Therefore, it cannot disentangle symptomatic improvements attributable to the treatment and naturally occurring fluctuations in symptoms. Furthermore, the short-term outcomes reported in the present study should be distinguished from long-term outcomes. As treatments may still be ongoing, remission rates after treatment termination will probably be higher. Therefore, comparisons to the remission rates of clinical trials, which typically report remission rates after treatment termination, are not feasible. Finally, it should be noted that the response rate of therapists, who provided information regarding treatment characteristics, was relatively low (67%). The strengths of the present study include the application of a thorough diagnostical procedure and well-validated measures among ED patients in nine different treatment facilities in Germany and Switzerland. It is one of the few studies evaluating outcomes of guideline-oriented ED treatment in routine clinical care.

In conclusion, the present study showed considerable symptomatic improvements among patients with AN and BN in routine clinical care. Among women with AN, full-time treatment was associated with substantial improvements in body weight, ED pathology, depressive symptoms, symptoms of anxiety, and somatic symptoms. Among women with BN, full-time treatment and ambulatory treatment were associated with considerable improvements on all measured variables, but with different treatment dosages. Results also show that full recovery of EDs is typically not achieved within three months of treatment initiation and requires prolonged treatment duration, even after initial symptomatic improvements during intensive hospitalization. Generally, results also show that treatment intensity and treatment outcomes are quite diverse, indicating the possibility for increasing effectiveness in the treatment of EDs in routine clinical care.

## Supporting information

**S1 Data. ED treatments in routine clinical care.**
(SAV)

## Author Contributions

**Conceptualization:** Silvia Schneider, Simone Munsch.

**Data curation:** Kathrin Schopf, Julia Lennertz, Nadine Humbel.

**Formal analysis:** Kathrin Schopf.

**Funding acquisition:** Silvia Schneider, Simone Munsch.

**Investigation:** Kathrin Schopf, Julia Lennertz, Nadine Humbel.

**Project administration:** Kathrin Schopf, Julia Lennertz, Nadine Humbel.

**Supervision:** Silvia Schneider, Andrea Hans Meyer, Nadine-Messerli Bürgy, Esther Biedert, Dirk Adolph, Simone Munsch.

**Validation:** Simone Munsch.

**Writing – original draft:** Kathrin Schopf.

**Writing – review & editing:** Kathrin Schopf, Silvia Schneider, Andrea Hans Meyer, Nadine-Messerli Bürgy, Andrea Wyssen, Esther Biedert, Bettina Isenschmid, Gabriella Milos, Malte Claussen, Stephan Trier, Katherina Whinyates, Dirk Adolph, Tobias Teismann, Jürgen Margraf, Hans-Jörg Assion, Bianca Überberg, Georg Juckel, Judith Müller, Benedikt Klauke, Simone Munsch.

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
