## [Decision Letter · Decision Letter 0]

2 Aug 2022

PONE-D-22-14198Implementation of eating disorder treatment in routine clinical care: A multi-center study examining short-term treatment outcomes among patients with anorexia nervosa and bulimia nervosa in Germany and SwitzerlandPLOS ONE

Dear Dr. Schopf,

Thank you for submitting your manuscript to PLOS ONE. After careful consideration, we feel that it has merit but does not fully meet PLOS ONE’s publication criteria as it currently stands. Therefore, we invite you to submit a revised version of the manuscript that addresses the points raised during the review process.

You will see from the accompanying comments that there was interest in your study, but the reviewers raised some important concerns too. I'm particularly concerned about the sample size issues and the data analysis strategy chosen, as suggested by one of the reviewers independent t-tests might not be the appropriate analyzes for comparing 3 groups on several variables. Please also specify the different types of treatment settings, whether in-patient, outpatient, or residential throughout the manuscript.

We look forward to receiving your revised manuscript.

Kind regards,

César González-Blanch, PhD

Academic Editor

PLOS ONE

Journal Requirements:

“We acknowledge support by the Open Access Publication Funds of the Ruhr-Universität Bochum.”

“This work was supported by the German Research Foundation (recipient: SSc, Grant SCHN 415/4-1, www.dfg.de) the Swiss National Science Foundation (recipient: SM, Grant 100013:149416, www.snf.ch) and the Swiss Anorexia Nervosa Foundation (recipient: SM, Grant 22-12, www.anorexia-nervosa.ch). None of the funders had a role in the study design, collection, analysis, or interpretation of the data, writing the manuscript, or the decision to submit the paper for publication.”

Reviewers' comments:

Reviewer's Responses to Questions

**Comments to the Author**

1. Is the manuscript technically sound, and do the data support the conclusions?

Reviewer #1: Partly

Reviewer #2: Partly

Reviewer #3: Partly

2. Has the statistical analysis been performed appropriately and rigorously? 

Reviewer #1: No

Reviewer #2: No

Reviewer #3: Yes

3. Have the authors made all data underlying the findings in their manuscript fully available?

Reviewer #1: Yes

Reviewer #2: Yes

Reviewer #3: Yes

4. Is the manuscript presented in an intelligible fashion and written in standard English?

Reviewer #1: Yes

Reviewer #2: Yes

Reviewer #3: Yes

5. Review Comments to the Author

Reviewer #1: This is basically a descriptive study which according to the investigators showed considerable symptomatic improvements among patients with AN and BN in routine clinical care. A highly-intensive, specialized residential treatment program was associated with large improvements within a short amount of time among women with AN. Inpatient and outpatient treatment were associated with almost comparable improvements among women with BN, but with different treatment dosages. Generally, results show that full recovery of EDs is typically not achieved within three months of treatment initiation and requires prolonged treatment duration, even after

initial symptomatic improvements. Generally, results also show that treatment intensity and treatment outcomes are quite diverse, indicating the possibility for increasing effectiveness in the treatment of EDs in routine clinical care.

As it is basically a descriptive study it appears that an inferential attempt at comparisons was really not the intent. However, the authors should explain the rationale for the sample size and its adequacy statistically for the findings, especially in the presentations of the Cohen’s d statistic.

Reviewer #2: The study compares outcomes in different settings of routine clinical care for patients with EDs.

One of the strengths of the study is that outcomes in all available settings of the German and Swiss health care system were included in one paper. Furthermore, authors recorded interviews and a part was coded by two raters which reflects high quality. Besides, I like that the authors assessed treatment components and treatment goals.

One of the main limitations of the study is that sample size per setting and per diagnosis is very low.

Introduction:

• The authors might replace references 2-5 by newer papers.

Methods:

• Participants and procedure: “approximately three months later”: do the authors have detailed information on this timepoint (mean, SD, range after admission)?

• Definition remission: I am wondering to which reference the authors refer to regarding e.g. the BMI=18 and EDE-Q<2.3 criteria? References 9, 37 and 38 seem to use other definitions? Furthermore, I am more familiar with a BMI=18.5 as remission criteria (and an EDE < 2.77). The authors should check if they used the most appropriate and most common remission criteria and might redo the remission analyses.

• Please specify how many of the 10 treatment facilities were inpatient, residential and outpatient.

• Statistical analyses: independent t-tests might not be the appropriate analyses comparing 3 groups on several variables. The authors might rather do MANOVAs with post-hoc-tests.

Results:

• Table 1: Is there any reason why n=114 is specified regarding age, but there is no information on “n” regarding the other variables

• Table 1: Do the authors have more detailed information regarding comorbidity?

• Table 2: The authors should specify ED severity and name the refence.

• Table 2: MMD is not mentioned in the table, please clarify.

Discussion:

• The authors should include as a limitation that response rates of therapists was only 67%.

• It seems that only 1 residential treatment center was included, so representativeness of this setting might be limited. The authors should mentioned this under limitations.

• BMI of inpatients seems rather high (BMI=16.5 vs. BMI<15 being often one of the criteria for inpatient treatment). Please discuss.

• From my point of view, the sample is not a strength of the study, but a limitation.

References:

• References 42 and 22 seem to be the same.

Reviewer #3: This is an interesting and generally well-presented paper. A few aspects were unclear to me, and these are noted below.

The title seems misleading in that it implies that effectiveness or efficacy the treatments will be examined, but in the abstract it's apparent that the study is more of a survey of existing treatments.

In the abstract, the purpose of the study is not entirely clear to me. Is the study designed to compare treatments in different clinics? Is it designed to determine whether treatments are effective?

In the introduction, the purpose of the paper needs to be stated explicitly.

In the methods, more info about the purpose of the larger trial will be of benefit.

A consort diagram for this study is required for transparency.

In the results section, when making comparisons, it's important to always state the comparator. For example, what the end of page 10, the comparator is unclear.

Throughout the paper, it could be helpful to make a more explicit definition of all the different types of treatment settings, whether in-patient, outpatient, or residential. I am not familiar with these terms, although I am familiar with the biology of eating disorders.

6. PLOS authors have the option to publish the peer review history of their article (what does this mean?). If published, this will include your full peer review and any attached files.

Reviewer #1: No

Reviewer #2: No

Reviewer #3: **Yes: **Amanda Sainsbury-Salis

---

## [Author Response · Author response to Decision Letter 0]

27 Sep 2022

Reviewers' comments:

Reviewer #1: 

This is basically a descriptive study which according to the investigators showed considerable symptomatic improvements among patients with AN and BN in routine clinical care. A highly-intensive, specialized residential treatment program was associated with large improvements within a short amount of time among women with AN. Inpatient and outpatient treatment were associated with almost comparable improvements among women with BN, but with different treatment dosages. Generally, results show that full recovery of EDs is typically not achieved within three months of treatment initiation and requires prolonged treatment duration, even after initial symptomatic improvements. Generally, results also show that treatment intensity and treatment outcomes are quite diverse, indicating the possibility for increasing effectiveness in the treatment of EDs in routine clinical care.

As it is basically a descriptive study it appears that an inferential attempt at comparisons was really not the intent. However, the authors should explain the rationale for the sample size and its adequacy statistically for the findings, especially in the presentations of the Cohen’s d statistic.

Response: It is correct that data of a randomized controlled clinical trial were subjected to secondary analyses for the present study. For the original clinical trial, an apriori power analysis (see study protocol, Munsch, 2014) was conducted. The power analysis referred to comparisons between clinical and control groups, but did not include comparisons within clinical subsamples. We agree with the reviewer that the sample sizes of subgroups are relatively small, which may limit generalizability of the present findings. In response, we have revised out methodological and statistical approach in order to avoid very low sample sizes and improve validity of findings. First, we have revised the comparison between treatment settings. In the revised manuscript, we only compare two settings (ambulatory and full-time treatment) instead of three settings (inpatient, outpatient, residential treatment). Furthermore, we added treatment outcomes for the total groups of patients with AN (n=59) and BN (n=57), as this more reliably reflects the average amount of change in patients (e.g., Table 5 and 6). In addition, we added regression analyses examining the relation between treatment intensity (number of psychotherapeutic sessions) and primary ED-related outcomes (weight gain and ED pathology) to better understand differences between full-time treatment and ambulatory treatment (p. 7, 12, 19). As the present study is one of the few studies evaluating symptomatic changes among patients with EDs in routine clinical care, we believe that this descriptive study contributes to this field of research. 

Reviewer #2: 

The study compares outcomes in different settings of routine clinical care for patients with EDs.

One of the strengths of the study is that outcomes in all available settings of the German and Swiss health care system were included in one paper. Furthermore, authors recorded interviews and a part was coded by two raters which reflects high quality. Besides, I like that the authors assessed treatment components and treatment goals. One of the main limitations of the study is that sample size per setting and per diagnosis is very low.

Response: We appreciate the reviewer’s positive appraisal of these study features and agree with his or her evaluation regarding the sample size. We refer to our response to Reviewer #1.

Introduction:

• The authors might replace references 2-5 by newer papers.

Response: We like to thank the reviewer for this suggestion and have updated the references (p. 27). 

Methods:

• Participants and procedure: “approximately three months later”: do the authors have detailed information on this timepoint (mean, SD, range after admission)?

Response: The post-measurement was scheduled to take place three months after study entry. As with other data collections, delays may have occurred due to personal reasons of participants (e.g., rescheduling of appointments). However, this happened only in a minority of instances. In response to the reviewer’s question, we changed the wording from “approximately three months” into “three months” throughout the manuscript.

• Definition remission: I am wondering to which reference the authors refer to regarding e.g. the BMI=18 and EDE-Q<2.3 criteria? References 9, 37 and 38 seem to use other definitions? Furthermore, I am more familiar with a BMI=18.5 as remission criteria (and an EDE < 2.77). The authors should check if they used the most appropriate and most common remission criteria and might redo the remission analyses.

Response: As suggested by the reviewer, we have updated the remissions criteria and applied the more commonly used criteria of BMI ≥ 18.5 and EDE-Q ≤ 2.77 (cf. Schmidt et al., 2015; DeJong et al. 2020). The methods section and the results section were revised accordingly (p. 10-11 and 19-20). 

• Please specify how many of the 10 treatment facilities were inpatient, residential and outpatient.

Response: We have added this information to the methods section (p. 8). 

Note: We also changed the number of cooperating treatment sites from ten to nine (as one of the cooperating sites did not treat ED patients). 

• Statistical analyses: independent t-tests might not be the appropriate analyses comparing 3 groups on several variables. The authors might rather do MANOVAs with post-hoc-tests.

Response: We like to thank the reviewer for this suggestion. When comparing groups on several related continuous variables, we conducted a multivariate analysis instead of several t-tests, as suggested (p. 11-12). The strategy for analyses and the results were revised accordingly. (Please note that, in the revised version of the manuscript, only comparisons between two treatment settings were conducted, due to changes in the methodological approach, as explained above). 

Results:

• Table 1: Is there any reason why n=114 is specified regarding age, but there is no information on “n” regarding the other variables

Response: We apologize for the confusion. This is just an error in consistency. We have removed this information from Table 1.

• Table 1: Do the authors have more detailed information regarding comorbidity?

Response: Yes, comorbid diagnoses were assessed. We have added more information regarding comorbidity to the sample descriptives (page 12). 

• Table 2: The authors should specify ED severity and name the reference. 

Response: ED severity specifiers (i.e., mild, moderate, severe, extreme) have been determined in the diagnostic interviews based on DSM-5 criteria. We added this specification to the manuscript (p. 9). 

• Table 2: MMD is not mentioned in the table, please clarify. 

Response: We apologize for the error, which pertains to an earlier version of the manuscript. We removed this information from Table 2.

Discussion:

• The authors should include as a limitation that response rates of therapists was only 67%. 

Response: As suggested, we added this information to the discussion of the limitations (p. 25). 

• It seems that only 1 residential treatment center was included, so representativeness of this setting might be limited. The authors should mentioned this under limitations. 

Response: We agree with the reviewer. To improve validity of the findings, we have revised our methodological and statistical approach. In the revised version of the manuscript, the residential treatment center is now being subsumed into “full-time treatment”. 

• BMI of inpatients seems rather high (BMI=16.5 vs. BMI<15 being often one of the criteria for inpatient treatment). Please discuss. 

Response: We agree with the reviewer that this is noteworthy and added a brief discussion of the differences between patients receiving ambulatory treatment setting and full-time treatment (including BMI) to the manuscript (p. 22-23). (Please note that, in the revised manuscript, the average BMI of inpatients with AN is 16.7, due to changes in the statistical analysis). 

• From my point of view, the sample is not a strength of the study, but a limitation. 

Response: We agree with the reviewer. In response to the reviewers’ remarks, we have revised our methodological approach with the aim to avoid very low sample sizes and improve validity of findings. We also revised the discussion of the strengths and limitations (p. 24-25). 

References:

• References 42 and 22 seem to be the same. 

Response: We like to thank the reviewer for this notification. This was corrected (p. 32).

Reviewer #3: 

This is an interesting and generally well-presented paper. A few aspects were unclear to me, and these are noted below.

The title seems misleading in that it implies that effectiveness or efficacy the treatments will be examined, but in the abstract it's apparent that the study is more of a survey of existing treatments.

Response: We agree with the reviewer and have changed the study title to better describe the research design. The new title is “Eating disorder treatment in routine clinical care: A descriptive study examining treatment characteristics and short-term treatment outcomes among patients with anorexia nervosa and bulimia nervosa in Germany and Switzerland”.

In the abstract, the purpose of the study is not entirely clear to me. Is the study designed to compare treatments in different clinics? Is it designed to determine whether treatments are effective?

Response: In response to the reviewers question, we have clarified the purpose of the study in the abstract. 

In the introduction, the purpose of the paper needs to be stated explicitly.

Response: The purpose of the study and the research questions were clarified in the introduction (p. 6-7). 

In the methods, more info about the purpose of the larger trial will be of benefit.

Response: As requested, we have added more information about the original clinical trial in the methods section (p. 7). 

A consort diagram for this study is required for transparency.

Response: The original consort diagram has been published in the main outcome paper of the randomized controlled trial (Munsch et al., 2021, Journal of Abnormal Psychology). We refer to this publication on page 7. As requested, we have also added a flow diagram (Fig 1), adapted for the present study. 

In the results section, when making comparisons, it's important to always state the comparator. For example, what the end of page 10, the comparator is unclear.

Response: We apologize for being unclear. We have revised the results section (p. 12-13) in order to more clearly state the comparator. 

Throughout the paper, it could be helpful to make a more explicit definition of all the different types of treatment settings, whether in-patient, outpatient, or residential. I am not familiar with these terms, although I am familiar with the biology of eating disorders.

Response: We have revised the description of treatment settings (p. 5) and now use the terms ambulatory treatment and full-time treatment throughout the manuscript.

---

## [Decision Letter · Decision Letter 1]

7 Nov 2022

PONE-D-22-14198R1Eating disorder treatment in routine clinical care: A descriptive study examining treatment characteristics and short-term treatment outcomes among patients with Anorexia Nervosa and Bulimia Nervosa in Germany and SwitzerlandPLOS ONE

Dear Dr. Schopf,

Thank you for submitting your manuscript to PLOS ONE. After careful consideration, we feel that it has merit but does not fully meet PLOS ONE’s publication criteria as it currently stands. Therefore, we invite you to submit a revised version of the manuscript that addresses the points raised during the review process. We are almost there. But one of the reviewers suggest a minor revision to your manuscript.  Therefore, you are invited to respond to the reviewer's comments and revise your manuscript. Please submit your revised manuscript by Dec 22 2022 11:59PM. If you will need more time than this to complete your revisions, please reply to this message or contact the journal office at plosone@plos.org. Please include the following items when submitting your revised manuscript:A rebuttal letter that responds to each point raised by the academic editor and reviewer(s). You should upload this letter as a separate file labeled 'Response to Reviewers'.A marked-up copy of your manuscript that highlights changes made to the original version. You should upload this as a separate file labeled 'Revised Manuscript with Track Changes'.An unmarked version of your revised paper without tracked changes. You should upload this as a separate file labeled 'Manuscript'.If applicable, we recommend that you deposit your laboratory protocols in protocols.io to enhance the reproducibility of your results. Protocols.io assigns your protocol its own identifier (DOI) so that it can be cited independently in the future. For instructions see: https://journals.plos.org/plosone/s/submission-guidelines#loc-laboratory-protocols. Additionally, PLOS ONE offers an option for publishing peer-reviewed Lab Protocol articles, which describe protocols hosted on protocols.io. Read more information on sharing protocols at https://plos.org/protocols?utm_medium=editorial-email&utm_source=authorletters&utm_campaign=protocols.

We look forward to receiving your revised manuscript.

Kind regards,

César González-Blanch, PhD

Academic Editor

PLOS ONE

Journal Requirements:

Reviewers' comments:

Reviewer's Responses to Questions

**Comments to the Author**

1. If the authors have adequately addressed your comments raised in a previous round of review and you feel that this manuscript is now acceptable for publication, you may indicate that here to bypass the “Comments to the Author” section, enter your conflict of interest statement in the “Confidential to Editor” section, and submit your "Accept" recommendation.

Reviewer #2: All comments have been addressed

Reviewer #3: All comments have been addressed

2. Is the manuscript technically sound, and do the data support the conclusions?

Reviewer #2: Yes

Reviewer #3: Yes

3. Has the statistical analysis been performed appropriately and rigorously? 

Reviewer #2: Yes

Reviewer #3: Yes

4. Have the authors made all data underlying the findings in their manuscript fully available?

Reviewer #2: Yes

Reviewer #3: Yes

5. Is the manuscript presented in an intelligible fashion and written in standard English?

Reviewer #2: Yes

Reviewer #3: Yes

6. Review Comments to the Author

Reviewer #2: The authors addressed all my concerns very well.

I have one minor comment regarding the revision:

I think it is confusing that Table 1 and Table 2 contain partly the same and partly different variables. The authors might also consider combining the information into 1 Table and report the 4 subgroups and the 2 total samples in 1 Table regarding all different variables from Table 1 and 2 (as in Table 5 and 6)

Reviewer #3: This revised paper is excellent, and all of my questions and concerns from previous Peer review has been addressed.

7. PLOS authors have the option to publish the peer review history of their article (what does this mean?). If published, this will include your full peer review and any attached files.

Reviewer #2: No

Reviewer #3: **Yes: **Amanda Sainsbury Salis

---

## [Author Response · Author response to Decision Letter 1]

20 Dec 2022

Reviewer #2: The authors addressed all my concerns very well. I have one minor comment regarding the revision:I think it is confusing that Table 1 and Table 2 contain partly the same and partly different variables. The authors might also consider combining the information into 1 Table and report the 4 subgroups and the 2 total samples in 1 Table regarding all different variables from Table 1 and 2 (as in Table 5 and 6)

Response: As suggested, we have combined Table 1 and 2 into one Table.

Reviewer #3: This revised paper is excellent, and all of my questions and concerns from previous Peer review has been addressed.

Response: We like to thank both reviewers very much for their time!

---

## [Editor Report · Decision Letter 2]

28 Dec 2022

Eating disorder treatment in routine clinical care: A descriptive study examining treatment characteristics and short-term treatment outcomes among patients with Anorexia Nervosa and Bulimia Nervosa in Germany and Switzerland

PONE-D-22-14198R2

Dear Dr. Schopf,

We’re pleased to inform you that your manuscript has been judged scientifically suitable for publication and will be formally accepted for publication once it meets all outstanding technical requirements.

Kind regards,

César González-Blanch, PhD

Academic Editor

PLOS ONE
---

## [Editor Report · Acceptance letter]

6 Jan 2023

PONE-D-22-14198R2 

Eating disorder treatment in routine clinical care: A descriptive study examining treatment characteristics and short-term treatment outcomes among patients with anorexia nervosa and bulimia nervosa in Germany and Switzerland 

Dear Dr. Schopf:

I'm pleased to inform you that your manuscript has been deemed suitable for publication in PLOS ONE. Congratulations! Your manuscript is now with our production department. 

Kind regards, 

on behalf of

Dr. César González-Blanch 

Academic Editor

PLOS ONE